# Unmet Needs for the Treatment of Chronic Hepatitis E Virus Infection in Immunocompromised Patients

**DOI:** 10.3390/v14102116

**Published:** 2022-09-25

**Authors:** Nassim Kamar, Arnaud Del Bello, Florence Abravanel, Qiuwei Pan, Jacques Izopet

**Affiliations:** 1Department of Nephrology, Dialysis and Organ Transplantation, CHU Rangueil, 31400 Toulouse, France; 2INSERM UMR 1291, Toulouse Institute for Infectious and Inflammatory Disease (Infinity), 31024 Toulouse, France; 3University Paul Sabatier, 31062 Toulouse, France; 4Laboratory of Virology, CHU Purpan, 31300 Toulouse, France; 5Department of Gastroenterology and Hepatology, Erasmus MC-University Medical Center, 3015 GD Rotterdam, The Netherlands; 6Erasmus MC Transplant Institute, Erasmus MC-University Medical Center, 3015 GD Rotterdam, The Netherlands

**Keywords:** hepatitis E virus, organ transplantation, ribavirin, treatment, needs

## Abstract

Hepatitis E virus (HEV) is the most prevalent hepatitis virus worldwide. Genotypes 3 (HEV3) and 4 (HEV4) as well as rat HEV can lead to chronic hepatitis E and cirrhosis in immunosuppressed patients. Within the last decade, several options for treating chronic hepatitis have been developed and have achieved a sustained virological response. However, there are still unmet needs such as optimizing immunosuppression to allow HEV clearance with or without ribavirin, as well as alternative therapies to ribavirin that are discussed in this paper.

## 1. Introduction

Hepatitis E virus (HEV) is the most prevalent hepatitis virus worldwide [1]. HEV is an RNA virus. There is one serotype and eight Genotypes 1 (HEV1) and 2 (HEV2), which are mainly prevalent in developing countries, are transmitted via the fecal-oral route [2]. Genotypes 3 (HEV3) and 4 (HEV4) are prevalent in developed countries, mostly in Europe and the United States of America, and have zoonotic transmission. HEV infection by genotypes 3 and 4 is a zoonosis. HEV3 is the most predominant genotype in Europe and the United States of America [2]. Emerging evidence suggests that Rocahepevirus ratti (rat HEV), belonging to the *Orthohepevirus C* species, can also infect humans [3,4].

The principal reservoirs of HEV3 are pigs, wild boar, mongooses, and rabbits [1]. HEV3 is transmitted mainly by the consumption of infected products (undercooked meat, infected crops or shellfish) or by the transfusion of infected blood products (red blood cells, platelets and plasma) [1,5]. In addition, HEV infection transmission transmitted by organ transplantation was also observed [6,7]. In the very large majority of cases, HEV3 infection is asymptomatic [8]. It can be responsible for acute hepatitis and jaundice, especially in patients having an underlying chronic liver disease, leading to so-called acute-on-chronic hepatitis [8], while genotypes 1 and 2 (HEV1 and HEV2, respectively) cannot evolve to a chronic form. Conversely, it is now well established that HEV3 and HEV4 as well as rat HEV can lead to chronic hepatitis and cirrhosis in immunosuppressed patients, i.e., solid-organ-transplant (SOT) patients, stem-cell-transplant patients, patients given chemotherapy for cancer or immunosuppressive therapy for auto-immune disease, as well as patients infected by the human immunodeficiency virus [9,10]. For instance, two-thirds of solid-organ-transplant patients will not clear the virus spontaneously within 3 months after the infection and will remain viremic and require, if possible, a reduction in immunosuppression and the use of anti-viral therapy [11,12]. Strikingly, nearly 10% of patients with persistent HEV replication, can develop cirrhosis within very few years [12]. Having a lymphopenia and receiving potent immunosuppressant agents such as tacrolimus (more potent than Cyclosporine A) were identified as risk factors for evolving to chronic hepatitis [12]. Within the last decade, several options for treating chronic hepatitis have been developed and have achieved a sustained virological response (SVR), defined by non-detection of HEV RNA in the serum and stools at least 3 months after ceasing therapy (Figure 1) [13]. However, there are still unmet needs that are discussed in this paper (Box 1).

Box 1List of unmet needs.            **Unmet Needs**
-Optimization of immunosuppression to allow HEV clearance in patients with chronic hepatitis;-Knowing how long HEV RNA should be undetectable in both serum and stools before stopping ribavirin;-Knowing how long relapsers should be retreated after a first course of ribavirin;-Optimization of immunosuppressant management under ribavirin therapy;-Prospective pharmacokinetic/pharmacodynamic studies to determine the optimal dose of ribavirin;-Determination of the mechanism of ribavirin action against HEV;-Management of patients with persisting HEV replication despite ribavirin;-Drug screening to identify new molecules that can stop HEV replication.


## 2. Decreasing Immunosuppression 

Due to the wide prevalence of HEV, patients with increased liver-enzymes levels should be screened systematically for HEV using approved serological tests and, if possible, nuclear acid assay tests [14,15,16]. In immunocompromised patients, the sensitivity of IgM serological tests is around 87% [14]. When HEV infection is suspected, HEV RNA should be looked for in the serum and/or in the stools even if anti-HEV IgM are negative [10]. Few studies showed that HEV antigen could be used to diagnose HEV infection in immunosuppressed patients (Soothil et al. Diagnostic utility of hepatitis E virus antigen-specific ELISA versus PCR testing in a cohort of post liver transplant patients in a large university hospital, Clin Virol. September 2018, 106, 44–48. doi: 10.1016/j.jcv.2018.07.007). In HEV RNA positive patients, it is recommended to reduce immunosuppression (when possible) and to wait for three months before initiating anti-viral therapy [2]. Indeed, it has been shown that one-third of SOT patients infected by HEV will spontaneously clear the virus [11]. These data were obtained in SOT patients who did not undergo a reduction in immunosuppression during the first 3 months after infection [11]. However, in patients who develop chronic hepatitis, defined by the persistence of HEV replication at 3 months after infection, decreasing immunosuppression, when possible, is considered a first line therapeutic option [11]. It achieves an SVR in one-third of patients with chronic hepatitis. Since an anti-HEV T-cell response is required to clear the virus, it is recommended to decrease immunosuppressants specifically targeting T-cells, namely, the calcineurin inhibitors. Indeed, it has been shown that patients who developed chronic hepatitis and who cleared the virus after a modification of immunosuppression had a lower tacrolimus median (min–max) trough level (3.25 (2.5–6.5) ng/mL) compared with those who remained viremic (7.35 (3.8–11.2) ng/mL), *p =* 0.02 [11]. Median daily steroid doses were also lower in patients who cleared the virus (0.035 (0.03–0.04) mg/kg compared to those who did not (0.1 (0.06–0.1) mg/kg, *p* = 0.04 [11]. Nevertheless, this is not feasible in all patients, particularly in patients at high immunological risk for acute rejection. Even in patients at low immunological risk, it is not clearly defined to what degree immunosuppressants should be reduced and what is the optimal level of immunosuppression to allow HEV clearance without over-exposing patients to a risk of acute rejection after solid-organ transplantation. It has been recently shown that in patients with chronic infection, the HEV-specific CD8+ T-cell response was diminished, declined over time, and displayed phenotypic features of exhaustion. However, improved proliferation of HEV-specific CD8+ T cells, increased interferon-γ production and evolution of a memory-like phenotype were observed upon reduction in immunosuppression and/or ribavirin treatment and were associated with viral clearance (Kemming et al., Mechanisms of CD8+ T-cell failure in chronic hepatitis E virus infection. J Hepatol. 2022 May 28; S0168-8278(22)00334-8. doi: 10.1016/j.jhep.2022.05.019). Monitoring specific anti-HEV T-cell response in immunosuppressed patients with chronic hepatitis could be helpful to determine the optimal threshold to obtain HEV clearance. Consequently, based on this threshold, immunosuppression can be tapered carefully to achieve HEV clearance.

## 3. Anti-Viral Therapy with Ribavirin

### 3.1. Ribavirin in SOT Patients

Ribavirin monotherapy achieved an SVR in up to 90% of SOT patients with chronic hepatitis E HEV infection [17,18]. Similar results were also reported in other immunocompromised patients such as stem-cell-transplant patients [19]. In most studies, ribavirin was given for a median duration of 3 months, ranging from 1 to 12 months. Initially, it was given empirically for 3 months. However, the persistence of HEV RNA in the stools, while no longer detected in the serum at 3 months after the initiation of ribavirin, was associated with increased risk of relapse after stopping therapy [20]. This prompted clinicians to look for HEV RNA in both serum and stools before stopping ribavirin [15]. Our group has shown that prolonging ribavirin therapy for as long as HEV RNA is still detected in the stools allows for a significant reduction in the risk of relapse after stopping therapy [21]. However, it is still unknown how long HEV RNA should be undetectable in both serum and stools before stopping ribavirin. In the event of relapse, patients can be retreated for a longer period [2,15,16]. Indeed, in retrospective studies, patients who relapsed after 3 months ribavirin therapy were retreated for a longer period [15,16]. The retreatment period ranged from 6 to 18 months [15,16]. This allowed researchers to obtain SVR in the large majority of cases (≈55%). However, it is still unknown how long this period should be.

Patients who achieved an SVR after ribavirin therapy had a higher lymphocyte count at the initiation of therapy compared with those who did not [18]. Hence, the management of immunosuppressants under ribavirin therapy should be studied. Some immunosuppressants such as mycophenolic acid can worsen hematological tolerance of ribavirin. Reducing ribavirin dose and requiring blood transfusion due to poor hematological tolerance were independent predictive factors for non-clearance of HEV after therapy [18]. As mentioned above, measuring specific anti-HEV T-cell response in this setting could improve the management of immunosuppression to optimize the response to ribavirin.

Both ribavirin and mycophenolic acid inhibit GTP pools via the inhibition of inosine monophosphate dehydrogenase [22]. In vitro, studies have suggested that ribavirin and mycophenolic acid have a synergistic effect against HEV [23]. However, this was not confirmed in vivo [24]. Conversely, the combination of both increases the risk of anemia. Hence, when possible, in case of persistent anemia, MPA mycophenolic acid could be stopped to allow ribavirin to be maintained at the optimal dose to achieve an SVR. It is important to note that ribavirin is contraindicated in pregnant women because it has teratogenic and embryonical effects.

The optimal doses of ribavirin are also unknown. As previously described in patients infected by hepatitis C, it is recommended to start with a dose that is adapted to kidney function to avoid ribavirin-induced anemia (Kamar et al. Ribavirin pharmacokinetics in renal and liver transplant patients: evidence that it depends on renal function. Am J Kidney Dis. 2004 Jan; 43(1):140–6. doi: 10.1053/j.ajkd.2003.09.019). Retrospective studies found no relationship between ribavirin trough levels and SVR [24], while others found that RBV plasma concentrations at steady state were significantly higher in clinical responders compared with clinical non-responders: median 1.96 (IQR 1.81–2.70) versus 0.49 (IQR 0.45–0.73) mg/L, *p* = 0.0004 [25]. Prospective pharmacokinetic/pharmacodynamic studies are required to determine the optimal dose of ribavirin to treat chronic hepatitis E HEV.

The mechanism of action of ribavirin against HEV should also be clearly determined. It has been suggested that ribavirin inhibits HEV replication via the depletion of GTP pools [22]. Another mechanism of action has also been proposed. In vivo, Ribavirin increases HEV quasispecies heterogeneity (mutagenesis) that seems to be reversible [26]. Other studies have shown that the presence of pretreatment HEV RNA polymerase mutations or the occurrence of HEV RNA polymerase mutations under ribavirin therapy are not associated with the virological response to ribavirin [18]. A total of 19 out of 76 transplant patients having pretreatment HEV RNA polymerase mutations failed to achieve SVR after a first course of ribavirin (25%) [18]. Hence, there is no robust data regarding the mechanism of action of ribavirin in the setting of HEV infection.

### 3.2. Ribavirin in Immunocompromised Non-SOT Patients

Ribavirin was also successfully used in other immunocompromised patients such as stem-cell-transplant patients [19]. Seventy-five percent of stem-cell-transplant patients given ribavirin monotherapy achieved SVR [19]. In HIV-infected patients, ribavirin also cleared HEV replication (Chronic hepatitis E in HIV patients: rapid progression to cirrhosis and response to oral ribavirin; Neukam K et al. Clin Infect Dis. 2013 Aug; 57(3):465–8. doi: 10.1093/cid/cit224. Epub 2013 Apr 10).

## 4. Alternative Therapies to Ribavirin

Although ribavirin is highly efficient in treating chronic hepatitis E HEV infection, a proportion of patients (≈10%) are either non-responders or remain viremic despite long-term ribavirin treatment. In these patients, alternative therapies could be considered. In very few case reports, pegylated interferon alone or in combination with ribavirin achieved an SVR [27,28,29]. Nevertheless, due to its immunostimulatory properties that can induce acute rejection [30], interferon cannot be used in all transplant patients, particularly in heart-, lung-, pancreas- and kidney-transplant patients. The combination of pegylated interferon and ribavirin was previously used for treating chronic hepatitis C virus infection in kidney-transplant patients without inducing an increased risk of acute rejection or impaired kidney function [31]. This combination was successfully used in HIV patients (Dalton, Treatment of chronic hepatitis E in a patient with HIV infection, Ann Intern Med. 2011 Oct 4; 155(7):479–80). However, this combination was not tested for treating chronic HEV infection in non-liver-transplant patients. It has been shown that sofosbuvir inhibits HEV replication in vitro and has a synergistic effect when combined with ribavirin [32]. However, in patients with chronic HEV infection who had previously experienced ribavirin failure, sofosbuvir failed to eliminate HEV RNA [33]. Similarly, in vitro, zinc salts blocked HEV replication by inhibiting the activity of viral RNA-dependent RNA polymerase [34]. Zinc also had a synergistic anti-HEV effect when combined with ribavirin [34]. Again, transplant patients with chronic HEV infection failed to respond to ribavirin despite high intra-erythrocyte zinc levels [35]. Silvestrol, a natural compound isolated from the plant Aglaia foveolate, inhibits HEV replication in vitro and in infected mice (Todt et al, The natural compound silvestrol inhibits hepatitis E virus (HEV) replication in vitro and in vivo. Antiviral Res. 2018 Sep; 157:151–158. doi: 10.1016/j.antiviral.2018.07.010). More recently, it was shown in vitro that niclosamide inhibits hepatitis E virus through suppression of NF-kappaB signaling [36]. By means of a screening platform, isocotoin was identified as candidate drug for targeting HEV replication. It inhibits HEV replication through interference with heat shock protein 90, a host factor not previously known to be involved in HEV replication [37]. However, niclosamide and isocotoin have not been tested in vivo. Very recently, 3 amino-rocaglates that possessed anti-viral activity against HEV were identified (Praditya et al. Identification of structurally re-engineered rocaglates as inhibitors against hepatitis E virus replication. Antiviral Res. 2022 Aug; 204:105359. doi: 10.1016/j.antiviral.2022.105359). None of them were tested in vivo. Hence, further studies are required to identify anti-viral therapies acting against HEV. Meanwhile, there is no recommendation for patients with persisting HEV replication despite ribavirin therapy. Initial studies have shown that liver fibrosis progresses rapidly in patients with chronic hepatitis E [11,12]. In a single report, liver fibrosis regressed in a liver-transplant patient with chronic hepatitis who continued ribavirin treatment for several years, although he remained viremic during therapy [38]. It is unknown whether continuation of ribavirin can be useful in this setting and have an anti-fibrotic effect. This should be determined. 

## 5. Anti-HEV Therapy for HEV Associated Extra-Hepatic Manifestations

It has been shown that HEV can also be responsible for extra-hepatic manifestations [39]. Clear relationships were established between HEV infection and neurological manifestations and kidney injuries. Guillain Barre syndrome, Neuralgic amyotrophy, encephalitis and peripheral neuropathy are the main manifestations that were reported at acute and chronic phase of HEV infection, in immunocompromised and non-immunocompromised patients [40,41,42,43]. HEV RNA was detected in cerebrospinal fluid (CSF) of transplant patients infected by HEV [44]. Interestingly, sequencing HEV strains showed that HEV variants could be different in the serum and the CSF suggesting that HEV neurotropic variants might exist [45]. There is no established therapy for HEV-induced neurological manifestations. Several empiric therapies were used: the reduction in immunosuppression, anti-viral therapies by ribavirin and/or interferon, intravenous immunoglobulins, and plasmapheresis [44]. The most adequate therapy is unknown, especially as it is unknown whether using ribavirin is useful in patients who become non-viremic but develop neurological manifestations. One can speculate that neurological manifestations are due to the immune response induced by HEV infection. In this case, using ribavirin in non-viremic patients is useless. Nevertheless, further studies are required to address this issue.

Glomerular diseases such as membranoproliferative glomerulopnephritis and membranous nephropathy were observed in transplant and non-transplant patients when infected by HEV [46,47]. Similarly to what had been observed in patients infected by the hepatitis C virus, some of these patients had cryoglobulinemia, and HEV RNA was detected in the cryoprecipitate [46]. Ribavirin allowed clearing HEV as well as cryglobulinemia [48]. Consequently, nephrotic syndrome regressed, and kidney function improved [48].

Some other manifestations, such as hematological disorders (thrombocytopenia and aplastic anemia) [49,50,51], pancreatitis [52], arthritis [53], and myositis [54] were reported in patients infected by HEV. However, no robust relationship was established between both diseases.

## 6. Conclusions

Ribavirin is very efficient for treating chronic HEV infection in immunocompromised patients. However, additional studies are required to understand its mechanism of action and to optimize its efficacy. Furthermore, drug screening should be carried out to identify other molecules that can stop HEV replication and that can be used in non-responders to ribavirin.

## Figures and Tables

**Figure 1 viruses-14-02116-f001:**
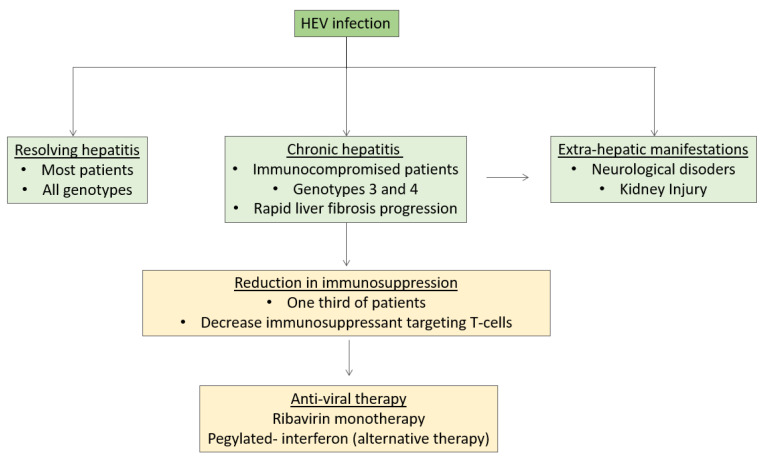
Natural history and management of hepatitis E.

## Data Availability

Not applicable.

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
