# Peer review of "Unmet Needs for the Treatment of Chronic Hepatitis E Virus Infection in Immunocompromised Patients"

_viruses, 2022, doi:10.3390/v14102116_

Round 1
Reviewer 1 Report
The authors provided a perspective that discusses chronic hepatitis E virus infections and cirrhosis in immunosuppressed patients. They illustrated the treatment options, their mechanism of action and efficacies. Although most of the data are mentioned in ESAL guidelines and in a previous publication by the same group, the focus on ttt in immunosuppressed patients is interesting and adds to the published data related to HEV. However, I have the following concerns that should be addressed:
General comments:
can you provide prospective for the treatment of HEV-infected pregnant women?
Recent publications reported high activity of other drugs against HEV but were not included in the paper. For example, https://pubmed.ncbi.nlm.nih.gov/30036559/.
Please include the effects of ribavirin on the different body compartments
What about the effect of using ribavirin in combination with other drugs for the treatment of HEV in immunocompromised patients?
Specific comments:
Introduction: authors refer to 4 HEV genotypes. However, the recent literature refer to 8 genotypes Please update the paragraph (https://www.ncbi.nlm.nih.gov/pmc/articles/PMC7660235/ )
It will be interesting if the authors added the mortality rate associated with chronic HEV infection
Figure 1 is mentioned in the manuscript text but the figure legend is missing
Under the title “2. Decreasing immunosuppression”, the authors described the role of IgM and NA only in the detection and diagnosis of HEV. Please illustrate the role of detecting HEV Ag in diagnosis.
Line 59: “In HEV RNA positive patients, it is recommended to wait for three months before initiating anti-viral therapy [2].”: Please illustrate!
The numbers on lines 70-73 are not clear. Are these the means, IQR, SD or else…..
Line 97: specify the frequency of cases that achieved SVR.
Line 111: the abbreviation of MPA is not shown in the manuscript.
I know that the paper focuses on immunocompromised patients, however, it will be interesting to compare the reported guidelines for ttt of acute HEV in immunocompetent cases (EASL) vs chronically immunosuppressed patients.
Line 113: is there a recommended dose for treatment?
Please discuss the treatment of HEV in SOT patients and in other immunocompromised (non-SOT) in separate titles to avoid confusion
Line 122: how frequent are these mutations associated with ttt failure?
Line 156: if no data is available supporting the anti-fibrotic effect of ribavirin, please delete.
Line 178: the sentence is not clear. “similarly,….” Please correct
Author Response
The here presented perspective article by Nassim Kamar and colleagues addresses the “Unmet needs for the treatment of chronic hepatitis E virus infection in immunocompromised patients”. All authors are very renown in the field and their work is a nice overview about current treatment options and blind spots in the clinical management of HEV infections.
We thank the reviewer for his nice comments
I only have a few minor comments that should be considered by the authors:
11) please use the new ICTV taxonomy for rat HEV species in line 27: Rocahepevirus ratti.
Done
22) the figure is lacking a legend; a treatment algorithm for chronic HEV infection should be more detailed and follow EASL guidelines.
Figure legend was added and the figure was modified.
33) start of line 56, do you mean “nucleic acid”?
Indeed, done
44) the statement in line 59 needs clarification. Guidelines recommend reduction of immunosuppression immediately after HEV RNA detection and only during this time no antiviral treatment is recommended.
This was clarified
55) the data on the role of T-cells in HEV infection is still controversially discussed. You could add information from a new paper out in JHepat (doi: 10.1016/j.jhep.2022.05.019). We have added this reference
66) it might be worth mentioning that most patients in the HepNet Pilot trial already experienced RBV treatment failure (line 140).
This was added
77) can you explain your focus in chapter 4? There’s data on silvestrol and other rocaglates to inhibit HEV replication in vitro and in vivo (10.1016/j.antiviral.2018.07.010; 10.1016/j.antiviral.2022.105359).
As requested, both references were added
88) I cannot quite follow chapter 5. The authors state, that there is no established therapy for HEV-induced extrahepatic manifestations (line 167), but isn’t this true for HEV per se? I suggest to carefully rephrase the paragraph.
I removed the sentence “there is no established therapy for HEV-induced extrahepatic manifestations”

Reviewer 2 Report
This article meticulously discusses the current knowledge on the strategies for treating chronic hepatitis E, most prevalent in Europe and the United States. My only concern is to revise the language to little extent in order to further enhance the readability and clarity of the subject. My suggested corrections/modifications are highlighted directly in the attached manuscript file.

Author Response
We thank the reviewer for his nice comments. The requested changes were done.

Reviewer 3 Report
The here presented perspective article by Nassim Kamar and colleagues addresses the “Unmet needs for the treatment of chronic hepatitis E virus infection in immunocompromised patients”. All authors are very renown in the field and their work is a nice overview about current treatment options and blind spots in the clinical management of HEV infections.
I only have a few minor comments that should be considered by the authors:
11) please use the new ICTV taxonomy for rat HEV species in line 27: Rocahepevirus ratti.
22) the figure is lacking a legend; a treatment algorithm for chronic HEV infection should be more detailed and follow EASL guidelines.
33) start of line 56, do you mean “nucleic acid”?
44) the statement in line 59 needs clarification. Guidelines recommend reduction of immunosuppression immediately after HEV RNA detection and only during this time no antiviral treatment is recommended.
55) the data on the role of T-cells in HEV infection is still controversially discussed. You could add information from a new paper out in JHepat (doi: 10.1016/j.jhep.2022.05.019).
66) it might be worth mentioning that most patients in the HepNet Pilot trial already experienced RBV treatment failure (line 140).
77) can you explain your focus in chapter 4? There’s data on silvestrol and other rocaglates to inhibit HEV replication in vitro and in vivo (10.1016/j.antiviral.2018.07.010; 10.1016/j.antiviral.2022.105359).
88) I cannot quite follow chapter 5. The authors state, that there is no established therapy for HEV-induced extrahepatic manifestations (line 167), but isn’t this true for HEV per se? I suggest to carefully rephrase the paragraph.
Author Response

(The authors gave the same response as above.)
